# Overexpression of the X-Linked Inhibitor of Apoptosis Protein (XIAP) in Neurons Improves Cell Survival and the Functional Outcome after Traumatic Spinal Cord Injury

**DOI:** 10.3390/ijms24032791

**Published:** 2023-02-01

**Authors:** David Reigada, Rodrigo M. Maza, Teresa Muñoz-Galdeano, María Asunción Barreda-Manso, Altea Soto, Dan Lindholm, Rosa Navarro-Ruíz, Manuel Nieto-Díaz

**Affiliations:** 1Molecular Neuroprotection Group, Research Unit, National Hospital for Paraplegics (SESCAM), 45071 Toledo, Spain; 2Department of Biochemistry and Developmental Biology, Faculty of Medicine, University of Helsinki, 00290 Helsinki, Finland; 3Minerva Foundation Institute for Medical Research, University of Helsinki, 00290 Helsinki, Finland

**Keywords:** spinal cord injury, X-linked inhibitor of apoptosis protein, neuroprotection, apoptosis, transgenic mice

## Abstract

Mechanical trauma to the spinal cord causes extensive neuronal death, contributing to the loss of sensory-motor and autonomic functions below the injury location. Apoptosis affects neurons after spinal cord injury (SCI) and is associated with increased caspase activity. Cleavage of X-linked inhibitor of apoptosis protein (XIAP) after SCI may contribute to this rise in caspase activity. Accordingly, we have shown that the elevation of XIAP resulted in increased neuronal survival after SCI and improved functional recovery. Therefore, we hypothesise that neuronal overexpression of XIAP can be neuroprotective after SCI with improved functional recovery. In line with this, studies of a transgenic mice with overexpression of XIAP in neurons revealed that higher levels of XIAP after spinal cord trauma favours neuronal survival, tissue preservation, and motor recovery after the spinal cord trauma. Using human SH-SY5Y cells overexpressing XIAP, we further showed that XIAP reduced caspase activity and apoptotic cell death after pro-apoptotic stimuli. In conclusion, this study shows that the levels of XIAP expression are an important factor for the outcome of spinal cord trauma and identifies XIAP as an important therapeutic target for alleviating the deleterious effects of SCI.

## 1. Introduction

Mechanical damage to the spinal cord results in a condition termed traumatic spinal cord injury (tSCI) which is characterised by partial or total loss of sensory-motor and autonomic functions below the injury location. It is one of the leading causes of disability in developed countries with big personal, economic, familiar, and social impacts [1,2,3]. Several promising therapies have been developed during the last three decades [3] and some are in different stages of clinical trials [3,4], but only early surgical decompression is routinely applied in clinical practice [5].

Mechanical trauma to the spinal cord primarily damages cell integrity leading to necrotic death among neural and non-neural cells. The processes initiated by the trauma trigger a secondary wave of damage, when different forms of programmed cell death (PCD) gradually replace necrosis [5,6,7,8,9,10,11], spreading the damage to remote areas of the central nervous system (CNS) [12]. All these PCD processes significantly contribute to the final extent of the injury and the loss of functions [7]. While prevention or modulation of necrotic death during acute SCI is unfeasible, preserving the surviving neural cells during secondary damage has been acknowledged as one of the main therapeutic targets for SCI [13]. As stated by Crowe and colleagues [6], the long-time course of the secondary injury represents a temporary window for treatment based on genetic and pharmacological regulation of programmed cell death pathways.

Among the different PCDs [14], apoptosis was the first to be identified in the damaged spinal cord, where it is considered a key feature of secondary damage [6]. Apoptotic cell death is controlled by the activation of the caspase cascade and cysteine-aspartyl proteases that are the last effectors of apoptosis and are activated in response to cell death stimuli (see review by Hanifeh and Ataei [15] and references therein). Molecular and biochemical data have shown that the members from the inhibitors of apoptosis (IAP) protein family regulate the apoptotic cell death program [16,17]. Among the members of this family, the X-linked inhibitor of apoptosis protein (XIAP) is the most potent and the most thoroughly characterised [15]. XIAP can directly bind and inhibit CASP3, 7, and 9, and also degrade CASP3 and 7 and other proteins that regulate apoptosis such as Smac [15]. In addition to apoptosis, XIAP also controls other cellular processes associated with trauma in the CNS [15].

XIAP cleavage correlated with the detection and cleavage of caspase 8, 9, and 3, leading the authors to hypothesise that caspase activation may be responsible for XIAP cleavage after SCI, “lowering the threshold of caspase activity necessary for inducing apoptosis”. In a later study [18], they demonstrated that the activation of NALP1 inflammasome in spinal neurons and other spinal cells results in XIAP cleavage as well as in the processing of IL-1β and IL-18 cytokines, contributing to tissue damage and functional deficits. In 2001, Keane and colleagues demonstrated the expression of XIAP in neurons and white matter cells of undamaged spinal cords and its cleavage during the first days after SCI [19] deficits. At the same time, studies using transgenic mice demonstrated that overexpressing human XIAP in neurons [20,21] resulted in increased survival and improved functional outcomes in models of cerebral hypoxia-ischemia [20,22] and amyotrophic lateral sclerosis [23].

Based on this, we have evaluated a pharmacological therapy based on ucf-101, a synthetic heterocyclic compound with neuroprotective effects in animal models of brain ischemia [24] that prevents XIAP cleavage and inhibition through the inhibition of Omi/HtrA2. Treatment with ucf-101 in a mouse model of contusive SCI resulted in reduced neuronal cell death, together with increased tissue sparing and improved functional outcomes. Results also suggested that the preservation of neuronal levels of XIAP after injury may be key to fostering their survival. However, indirect effects can be expected, since ucf-101 may also be neuroprotective through the activation of other pathways such as ERK1/2 [25,26,27], and its systemic administration may have protective effects in other cell types, which may indirectly contribute to protecting neurons.

The main aim of the present study was to confirm the protective role of increasing XIAP protein expression in neurons and to evaluate its effect on the amelioration of tissular and functional damage induced by SCI. To this aim, we assayed two models of genic overexpression of XIAP in neurons. First, was an in vivo model of contusive SCI in transgenic mice overexpressing the human XIAP protein (hXIAP) in postnatal neurons, which allowed us to evaluate the benefits of XIAP overexpression for neuronal survival and the consequences of neuroprotection for tissue preservation and functional recovery. Second, using an XIAP-overexpressing human neuroblastoma cell line (SH-SY5Y^XIAP+^) allowed us to identify the molecular pathways implied in the transgene effect on neuronal apoptotic death.

## 2. Results

### 2.1. Overexpression of hXIAP in the Spinal Cord of CB57BL/6 Mice (CB57^hXIAP+^)

Reduction in XIAP expression after SCI in CB57^WT^ mice (Figure 1A,B) can be an important factor for the secondary neuronal death induced by the trauma to the spinal cord. To test this possibility, we employed a contusive SCI model in Thy1-XIAP transgenic mice overexpressing human XIAP protein (CB57^hXIAP+^, see details in the methods section and in Trapp and colleagues, 2003 [20]).

As shown before [20,22,23,28], the Thy 1.2 expression cassette drives the overexpression of hXIAP in neurons in different parts of the CNS such as the hippocampus, the cortex, the striatum, and the spinal cord. To confirm that the construction induces hXIAP overexpression in the spinal cord and to analyse its changes after SCI, we carried out an immunoblot using a monoclonal antibody against hXIAP without known reactivity against mouse XIAP in spinal cord protein samples from sham (0 days post-operation (0 dpo)) and SCI (1 and 3 dpo) CB57^hXIAP+^ mice and from 0 dpo CB57^WT^ control mice (Figure 1C,D). As expected, hXIAP was highly expressed in the spinal cord of CB57^hXIAP+^ mice and absent in CB57^WT^. After SCI, hXIAP expression in transgenic mice became significantly reduced, although remained detectable (difference vs. sham CB57^hXIAP+^ mice according to Student’s *t*-test: 1 dpo: T_4_ = 2.84, *p*-value = 0.02; 3 dpo: T_5_ = 2.59, *p*-value = 0.03; *n* = 3 animals/group).

### 2.2. Overexpression of hXIAP Protects Neurons from SCI Secondary Damage

Spinal cord sections from CB57^WT^ and CB57^hXIAP+^ mice labelled for the NeuN neuronal marker were used to quantify the number of neurons surviving after injury. Estimates of the number of neurons in slices 4.8 mm around the injury epicentre showed that overexpression of hXIAP in spinal neurons from CB57^hXIAP+^ mice had neuroprotective effects (Figure 2A). These analyses revealed that at 21 dpo, virtually all neurons had disappeared within the 400 µm surrounding the injury epicenter in both CB57^WT^ and CB57^hXIAP^ animals (number of neurons in CB57^WT^ mice = 36.3 ± 35.1 vs. sham = 800 ± 39 and in CB57^hXIAP+^ mice = 32.75 ± 15.6 vs. sham = 807.7 ± 72.1). In CB57^WT^ mice, sections at 2 mm rostral and caudal to the epicentre retained about 50% of neurons observed in sham mice (2 mm caudal: 51.21 ± 10.17% reduction, T_6_ = 5.24, *p*-value = 0.001; 2 mm rostral: 43.32 ± 23.82% reduction, T_7_ = 3.93, *p*-value = 0.003; *n* = 3–6 animals/group). Conversely, neuronal loss was reduced in CB57^hXIAP+^ mice in sections 1 to 2 mm at both rostral and caudal directions compared with CB57^WT^ mice, reaching similar counts as in sham conditions (2 mm caudal: 27.49 ± 18.86% reduction in the number of neurons relative to sham CB57^hXIAP+^; T_6_ = 4.08, *p*-value = 0.003; *n* = 3–6 animals/group; 2 mm rostral: only a 21.02 ± 10.73% reduction; T_3_ = 3.23, *p*-value = 0.02).

### 2.3. XIAP Overexpressing Mice Have a Higher Degree of Tissue Preservation after Injury

To evaluate the effect of the overexpression of hXIAP on the amount of spared white matter area after SCI, we stained mouse spinal cord slices with eriochrome cyanine and analysed them using Cavalieri’s stereological method. White matter in sham CB57^WT^ and CB57^hXIAP+^ mice represented approximately 55% of the transversal section area of the spinal cord. In CB57^WT^ mice at 21 dpo, tissue damage reduced the spared white matter volume to approximately 30–25% of the section at the epicentre. As the distance to the epicentre increased, tissue preservation increased accordingly, showing control values at 1 mm at both caudal and rostral directions to the injury site (Figure 3A,B). Although no differences in tissue preservation between CB57^hXIAP+^ and CB57^WT^ mice were observed at the epicentre or rostral sections, overexpression of hXIAP in CB57^hXIAP+^ mice significantly increased white matter preservation in caudal sections close to the injury site (48.23 ± 1.31% white matter volume from 0.2 to 0.5 mm to the epicentre in CB57^hXIAP+^ mice vs. 37.75 ± 2.05% in CB57^WT^ mice; T_9_ = 3.57, *p*-value = 0.003, *n* = 5 animals per group) (Figure 3C), reaching values from sham animals much closer to the epicentre (500 µm) than in wild-type (WT) individuals.

Improvement in tissue preservation (Figure 3) and neuron counts (Figure 2) in injured CB57^hXIAP+^ compared with injured CB57^WT^ mice followed almost the same pattern, being both higher in caudal areas to the injury epicentre than in the epicentre or the rostral areas. As expected, tissue preservation data significantly correlated with the number of survival neurons at all levels (Figure 3D) (Pearson’s correlation coefficient (PCC) = 0.6, *p*-value = 0.012).

### 2.4. Overexpression of hXIAP Improves Locomotor Recovery

Evaluation of the locomotion function recovery using the BMS scale confirmed that the neuroprotective effects of hXIAP overexpression in CB57^hXIAP+^ mice led to a reduction in the motor skill deficits derived from SCI (Figure 4A). SCI caused a total loss in hindlimb locomotor ability, with a BMS score of 0 or 1, in the first two dpo that were spontaneously and partially recovered in the next 21 (BMS score = 5.2 ± 0.2) and 28 dpo (5.4 ± 0.2) in CB57^WT^ animals (*n* = 8–12 animals/group). The BMS test reached significantly higher scores in CB57^hXIAP+^ mice compared with CB57^WT^, reaching a mean BMS score of 5.7 ± 0.3 points at 21 dpo (*n* = 12) and 6.4 ± 0.3 at 28 dpo (*n* = 12), (F_1_ = 8.6; *p*-value = 0.009 for genetic background effect in a two-way ANOVA).

As illustrated in Figure 4B, the BMS subscore emphasised the differences between the CB57^WT^ animals (mean subscore 3.6 ± 0.7 at 21 dpo (*n* = 8) and 3.6 ± 0.8 at 28 dpo (*n* = 12)) and CB57^hXIAP+^ mice (4.9 ± 0.8 points at 21 dpo (*n* = 12) and 5.8 ± 0.6 at 28 dpo (*n* = 12)), (F_1_ = 8.5; *p*-value = 0.01 for genetic background effect in a two-way ANOVA). Differences were particularly obvious in the capacity of mice to carry out a coordinative walk between forelimbs and hindlimbs. An individualised analysis revealed that a higher number of CB57^hXIAP+^ mice recovered a mostly coordinated locomotion compared with CB57^WT^ mice (75% (six of eight animals) vs. 28% (two of seven) at 28 dpo, respectively; Chisquare_1_ = 3.23, *p*-value = 0.07; *n* = 7–8 animals/group) (Figure 4C).

As shown before with the relationship between tissue preservation and cell counts, we evaluated whether the improved motor performance observed in injured CB57^hXIAP+^ correlates with the improvements at tissue and cellular levels. Firstly, a correlation analysis showed that motor function recovery (BMS subscore) had a good correlation with neuron counts in rostral and caudal sections (Figure 5A) (epicenter: PCC = −0.5, *p*-value = 0.15; rostral: PCC = 0.78, *p*-value = 0.06; caudal: PCC = 0.56, *p*-value = 0.12).

Additionally, subscores were not correlated to rostral or epicentre levels (Figure 5B) (epicentre: PCC = 0.07, *p*-value = 0.41; rostral: PCC = 0.2, *p*-value = 0.27) but significantly correlated at caudal level (PCC = 0.51, *p*-value = 0.05).

### 2.5. Overexpression of hXIAP Reduces Cell Death in SH-SY5Y Cells

In vivo experiments suggested that hXIAP overexpression in neurons protected them from SCI-induced cell death, resulting in enhanced locomotor recovery. To explore the implication of the anti-apoptotic effects of XIAP in this observed improvement, we analysed the effects of overexpression of XIAP (Figure 6A) in cultures of the human neuroblastoma cell line SH-SY5Y (SH-SY5Y^XIAP+^). We compared the effects of pro-apoptotic stimuli staurosporine (Sts) and thapsigargin (TG) in comparison with WT cells (SH-SY5Y^pcDNA3^).

Flow cytometry of DAPI-stained SH-SY5Y cells showed that treatment with Sts for 24 h significantly increased the percentage of cells with condensed nuclei (sub-G_0_ stage indicated in Figure 6B and its insert) in a dose-dependent manner (Figure 6C). XIAP overexpression in SH-SY5Y^XIAP+^ cells reduced both the maximum percentage of cell death reached by increasing concentrations of Sts (E_max_) and the half maximal effective concentration (EC_50_) (SH-SY5Y^pcDNA3^: E_max_ = 87.75% and EC_50_ = 8.86 nM; SH-SY5Y^XIAP+^: E_max_ = 65.73% and EC_50_ = 13.81 nM), which was a significant reduction induced by the genetic background (F_1_ = 9.34; *p*-value = 0.03 in a two-way ANOVA).

TG treatment (2.5 µM for 24h) also induces death in SH-SY5Y^pcDNA3^ cells. We employed this treatment to evaluate the effect of XIAP overexpression on the advance of apoptosis. Initially, we evaluated the first phases of the cell death process or early apoptosis (Figure 6D), measuring the translocation of phosphatidylserine in the membrane through the annexin V staining assay followed by flow cytometry measurement. Results showed that TG treatment significantly increased the percentage of cells with annexin V staining (control = 13.9 ± 1.63%; TG-treated = 45.62 ± 6.49% (T_8_ = 4.75, *p*-value < 0.001 in a Student’s *t*-test; *n* = 5). Overexpression of XIAP in SH-SY5Y^XIAP+^ cells significantly reduced the number of cells stained for annexin V compared to SH-SY5Y^pcDNA3^ (control = 12.73 ± 1.63%; TG-treated = 22.4 ± 2.8%; F_1_ = 8.9, *p*-value = 0.009 for genetic background effect in a two-way ANOVA; TG-treated cells, SH-SY5Y^XIAP+^ vs. SH-SY5Y^pcDNA3^; T_7_ = 3.00, *p*-value < 0.001 in a Student’s *t*-test; *n* = 4). Finally, we measured a late phase of apoptosis (Figure 6E), detecting condensed nuclei by staining nucleic acids with DAPI dye followed by flow cytometry measurement.TG also increased the % of cells showing nuclear DAPI staining in SH-SY5Y^pcDNA3^ cells, signs of a late stage of apoptosis (control = 18.08 ± 7.9%; TG-treated = 40.5 ± 7.8%), which was strongly but not significantly reduced by XIAP overexpression in SH-SY5Y^XIAP+^ (control = 15.44 ± 2.7%; TG-treated = 22.4 ± 2.8%: F_1_ = 2.06, *p*-value = 0.17 for genetic background effect in a two-way ANOVA; TG-treated, SH-SY5Y^XIAP+^ vs. SH-SY5Y^pcDNA3^; T_7_ = 1.67, *p*-value = 0.07 in a Student’s *t*-test; *n* = 5).

### 2.6. Overexpression of XIAP Reduces Caspase Cleavage and Activity in SH-SY5Y Cell Cultures

Immunoblot analyses of protein samples from SH-SY5Y^pcDNA3^ cultures revealed that stimulation with 25 nM Sts for 24 h reduced the expression of XIAP and cleavage of CASP3. We also observed both basal and stimulated cleavage of PARP, a specific substrate of CASP3 activity. SH-SY5Y^XIAP+^ cells maintain high levels of XIAP even after Sts treatment, and reduced cleavage of pro-CASP3 protein after Sts stimuli. We also observed that both basal and stimulated CASP3-dependent cleavage of PAPR is reduced in SH-SY5Y^XIAP+^ cells (Figure 7). We did not observe any change in levels of pro-CASP7.

In agreement with the observed cleavage of pro-CASP3 protein, evaluation of caspase activity by the Caspase-Glo chemiluminescent assay revealed that Sts (25 nM for 24 h) induced a significant increase in CASP3/7 activity in SH-SY5Y^pcDNA3^ cells (5.33 ± 0.007-fold increase relative to non-stimulated levels after 30 min; T_2_ = 6.59, *p*-value = 0.011 in a Student’s *t*-test; *n* = 3) (Figure 8A,B). The overexpression of XIAP in SH-SY5Y^XIAP+^ cells almost abolished the effect of Sts on CASP3/7 activation (only a 2.82 ± 0.62-fold increase compared to non-stimulated levels after 30 min) and significantly reduced the effect of the treatment compared with SH-SY5Y^pcDNA3^-treated cells (T_2_ = 3.70, *p*-value < 0.033 vs. SH-SY5Y^pcDNA3^-treated cells in a Student’s *t*-test; *n* = 3).

These results demonstrate that locomotor improvements found after SCI were derived from the neuronal overexpression of XIAP and resulted, at least in part, from the inhibition of caspase cleavage by XIAP.

## 3. Discussion

Through this study, we have confirmed that overexpression of XIAP effectively reduces caspase activity and apoptotic death in neural cells exposed to deleterious stimuli, protecting spinal neurons from secondary damage in a mouse model of contusive SCI and contributing to tissue sparing and motor function recovery.

To determine the specific effects of preserving XIAP expression in neurons for the outcomes of SCI, we employed transgenic mice bearing a human XIAP transgene (CB57^hXIAP+^), which overexpress XIAP in the spinal cord [23] and other regions of the CNS such as the cortex, hippocampus, or striatum [20,22,28]. As observed with the endogenous XIAP in WT animals (see Figure 1, and Keane and colleagues, 2001 [19]), the levels of uncleaved hXIAP protein dropped in the transgenic mice during the first days after injury in CB57^hXIAP+^ mice. Similar reductions were observed after neonatal hypoxia-ischemia in these transgenic mice [22], which may result from XIAP cleavage by caspases [18,19], Omi/HtrA2 [25], calpains [29], or the inflammasome [30]—which are also activated or overexpressed after SCI—or due to the loss of neurons expressing XIAP after SCI.

Despite its reduction after injury, full-length hXIAP in the transgenic mice remained detectable in the damaged spine and its effects or those from its fragments [19] were functionally relevant, as revealed by a marked increase in the quantity of surviving neurons after SCI. Effects were marked in penumbra regions around the injury epicentre, with transgenic mice showing significant neuronal losses restricted to the 2 mm area surrounding the injury epicentre compared to above 5 mm in WT mice. Neuronal protection resembled that observed after ucf-101 treatment [25] in the extent and also in the spatial distribution, being more marked in the penumbra region caudal to the injury although transgenic mice show a significant increase in neuronal survival also in the rostral penumbra. Neuroprotection by hXIAP overexpression was also observed in the modified SH-SY5Y neuroblastoma cell line (SH-SY5Y^XIAP+^), which preserved enough functionally active XIAP under pro-apoptotic conditions to reduce caspase cleavage and activity, reduce PARP cleavage, and increase cell survival against Sts and TG stimulation. Altogether, the present results suggest that the levels of XIAP may be a crucial factor for neuronal survival after SCI and that XIAP, directly or indirectly, can interfere with the development of SCI secondary damage, as previously proposed for hypoxia, ischemia, and ALS [20,22,23].

Increased neuronal survival in the transgenic mice was significantly correlated with an increase in white matter preservation, particularly in the penumbra region caudal to the injury epicentre and to a lesser degree in the rostral regions, similar to our observations after ucf-101 treatment [25]. Increases in tissue preservation agree with the observations by Wang and colleagues [22] in the analysis of the effects of XIAP overexpression after cerebral ischemia using the same strain of transgenic mice. These results suggest that neuronal death significantly contributes to tissue degeneration during SCI secondary damage, similar to the demyelination and gliosis observed in ALS following motor neuron degeneration [31]. Further studies, including cultures of spinal neurons overexpressing XIAP or single cell analyses of CB57^hXIAP+^ mice, would be required to define the precise survival/cell death pathways that become regulated by XIAP overexpression in the specific neuronal populations of the spinal cord.

Increased tissue and neuronal preservation in the penumbra of the injury results can be expected to result in improved locomotion recovery [32]. In agreement, motor function in CB57^hXIAP+^ after SCI matches the functional improvements found after systemic ucf-101 treatment, reaching an also higher degree of interlimb coordination than in WT mice (CB57^WT^). Coordination is a major milestone that largely depends on local networks of propriospinal interneurons (see review by Flynn and colleagues, 2011 [33] and references therein), which it is only achieved by a very reduced number of untreated CB57BL/6 mice following a moderate thoracic contusion. It usually precedes noticeable recovery [34], contributing to functional recovery after SCI through the formation of new networks reconnecting supraspinal tracts with their spinal targets.

In summary, this study shows that the decrease in XIAP expression is an important factor in the effects of trauma and identifies XIAP as an important therapeutic target to foster neuronal survival and improve SCI outcome.

## 4. Materials and Methods

### 4.1. In Vivo

#### 4.1.1. Thy1-XIAP Transgenic Mice

Thy1-XIAP mice are a strain of transgenic mice that overexpress human XIAP in postnatal neurons without obvious phenotypic differences relative to WT in normal conditions [20]. The generation of Thy1-XIAP transgenic mice has been fully described by Trapp and colleagues(2003) [20]. Briefly, the human 1.5-kb XIAP cDNA was subcloned into the XhoI sites of the Thy 1.2 expression cassette, replacing the endogenous exon 3 and flanking sequences. It has been shown that the Thy 1.2 expression cassette drives transgene expression in postnatal neurons [20,35,36]. Transgenic mice (CB57^hXIAP+^) were generated by the injection of gel-purified DNA into fertilised oocytes. The eggs were transferred into the oviducts of pseudopregnant females by using standard techniques [37]. The genetic background in which the transgenic mice were made was CBA × CB57BL/6. Transgenic founder mice were identified by PCR using genomic DNA extracted from tail samples and amplifying the human XIAP gene using the primer described in Appendix A. Five mouse lines were mated further. Animals with the same genetic background but no transgene inserted served as controls (WT or CB57^WT^ mice).

#### 4.1.2. Spinal Cord Injury Procedure

WT (CB57^WT^) and CB57^hXIAP+^ transgenic C57BL/6 mice were housed in plastic cages in a temperature- and humidity-controlled room maintained with a 12:12 h reverse light/dark cycle with free access to food and water. All manipulations and treatments were carried out in full accordance with the guidelines on the care and management of animals established by the European Union (directive 86/609/CEE), the guidelines on the use of animals for Neuroscience Research of the Society for Neuroscience, the NIH guide for the care and use of laboratory animals, and the normative R.D. 1201/2005 10-10 from the Spanish Ministry of the Environment and the Agriculture Council of the Castilla-La Mancha animal ethics committees. All procedures were approved by the Hospital Nacional de Parapléjicos Animal Care and Use Committee (153BCEEA/2016). All efforts were made to minimise suffering as well as the number of animals used. Female mice of 20 g of weight (12–14 weeks of age) were anesthetised through isoflurane inhalation (2% in oxygen for induction and 1.5% during surgery, Forane, Baxter Healthcare Corporation, Deerfield, IL, USA). The spinal cord T8 segment was exposed by laminectomy in the 9th thoracic vertebrae (T9) and subsequently received a moderate 50Kdyne contusion using an IH Spinal Cord Impactor device (Precision System & Instrumentation, Lexington, KY, USA). After surgery, the mice were treated with analgesics 0.03 mg/kg buprenorphine; Buprex, Reckitt Benckiser Pharmaceuticals Limited, Richmond, VA, USA) and antibiotics (0.4 mg/kg enrofloxacin; Baytril, Bayer AG, Leverkusen, Germany), together with daily manual bladder emptying for 2 weeks. In the sham group, mice underwent a laminectomy without contusion and were maintained as the injured animals.

#### 4.1.3. Evaluation of hXIAP Expression Levels in Spinal Cord Tissue by Immunoblot

At the corresponding dpo, animals were euthanised by administration of a lethal dose of sodium pentobarbital (40 mg/kg Dolethal, Vetoquinol, Madrid, Spain), and a sample of 1.5–2 cm of the spinal cord around the injury site was extracted. Tissue samples were homogenised in radioimmunoprecipitation assay buffer (RIPA; Sigma-Aldrich, St Louis, MO, USA), supplemented with protease inhibitors (Complete Protease Inhibitor Cocktail Tablets, Roche, Bassel, Switzerland), sonicated, and cleared by centrifugation (10,000× *g* for 15 min at 4 °C). The homogenate containing 50 µg of protein was resolved using conventional sodium dodecyl sulphate-polyacrylamide gel electrophoresis (SDS–PAGE) in reducing conditions (5% β-mercaptoethanol; Sigma-Aldrich) and transferred to a polyvinylidene difluoride membrane (PVDF; Immobilon, Merck Millipore, Darmstadt, Germany). The membrane was blocked with a blocking solution of 5% nonfat milk in TBS-T (Tris buffer saline plus 0.05% (*v/v*) Tween20; Sigma Aldrich) for 1 h at room temperature (RT) and incubated with the corresponding anti-human XIAP antibody overnight at 4 °C, using β-actin as a loading control (see antibody list in Appendix A). Afterward, blots were incubated at RT for 1 h with the correspondent secondary antibody diluted in a blocking solution (1:5000). Membranes were developed by a SuperSignal West Pico chemiluminescent assay (Pierce, ThermoFisher Scientific, Waltham, MA, USA). All employed antibodies exhibited the specific band or bands of the expected molecular weight for their target/s but also a variable number of non-specific bands that were not considered.

#### 4.1.4. Histology

At 21 dpo, mice were euthanised with a lethal dosage of sodium pentobarbital and transcardially perfused with 25 mL of saline buffer with sodium heparin (1 unit/mL, Chiesi España, Barcelona, Spain) followed by 50 mL of 4% (*w/v*) paraformaldehyde in 0.1 M phosphate buffer at pH 7.4. A segment of 1 cm of spinal cord around the injured area was removed, embedded in OCT (Tissue-Tek; Sakura Finetek Europe B.V., Alphen aan den Rijn, The Netherlands), and frozen at −20 °C. Embedded tissue was cut into 20 µm sections using a cryostat (HM560, Microm International GmbH, Walldorf, Germany) and mounted on microscope glass slides (Superfrost Plus, ThermoFisher Scientific).

#### 4.1.5. Immunofluorescence and Neuronal Counting

Frozen spinal cord sections were first heated at 37 °C for 1 h, rehydrated in phosphate-buffered saline (PBS), and blocked and permeabilised by incubation for 2 h at RT in a blocking solution composed of 1% (*w/v*) bovine serum albumin (BSA, Sigma-Aldrich), 2% horse serum (Sigma-Aldrich), and 0.2% (*v/v*) Triton X-100 (Sigma-Aldrich) in PBS. Afterward, sections were incubated overnight at 4 °C with the specific neuronal marker rabbit anti-neuronal nuclei protein NeuN (1:500; see details in Appendix A) diluted in 0.1% BSA, 0.1% horse serum, and 0.2% TritonX-100 in PBS. Then, sections were incubated for 2 h at RT with an Alexa Fluor 488 nm conjugated goat anti-rabbit secondary antibody (1:500; see details in Appendix A) in 0.1% BSA, 0.1% horse serum, and 0.2% TritonX-100 in PBS. Finally, cell nuclei were stained with the fluorescent marker of nucleic acids,4′,6-diamino-2-fenilindol (DAPI 1:3000; Sigma-Aldrich), and mounted with a PermaFluor Mountant Medium (Thermo Scientific). Microscope inspection of the sections confirmed the specificity of the neuronal staining.

For neuron count, we acquired images of the whole mouse spinal cord sections through the macro to micro function of an IX83 scanR microscope (Olympus, Center Valley, PA, USA) using a 10× objective at high magnification and the deeplearning-based image analysis approach TruAI integrated CellSens Dimensions software (Olympus), that uses deep convolutional neural network architecture for object segmentation. During the training phase, we fed the network with nearly 1000 manually segmented neurons across the grey matter of control and injured spinal cords. The background data surrounding each neuron, as well as artifacts, were also identified. This training phase was carried out using the Deep Learning module operating under the Standard Network configuration and the Olympus protocols on 300,000 iterations with 5 checkpoints every 60,000 iterations. Although predictions by TruAI can be very precise and robust, the generated neural network was validated using the Olympus CellSens imaging software to ensure that no artifacts or other errors were produced. A minimum of 70% congruence between manual and neural network identifications was set.

Once the neural network had been trained and validated, it was applied to all images to assign the probability of being part of a neuronal nucleus to every pixel in the image. To identify neuronal nuclei, we considered only those particles composed of pixels above 50% probability and an area above 25 µm^2^. The designed neural network, as well as all identifications and every code employed in these identifications, are available at NeuroCluedo (https://osf.io/an7f2, accessed on 29 November 2022).

#### 4.1.6. Eriochrome Cyanine Stain of Myelin

To determine the area and volume of spared tissue, we followed a modified method of eriochrome cyanine (EC) staining from Rabchevsky and colleagues [38] and described in detail by Reigada, 2022 [39]. Briefly, sections were sequentially stained with EC followed by counter-staining with 5% iron alum and borax-ferricyanide solutions. After dehydration with increasing concentrations of ethanol and Histochoice Cleaning Agent (Sigma-Aldrich), the samples were mounted using DPX (Sigma-Aldrich). EC stained white matter myelin and allowed us to differentiate spared white matter from grey matter or damaged tissue. The area and volume of spared white matter were estimated through the stereological analysis of sections comprising 1 cm around the epicentre (200 µm between sections), using Cavalliery’s method in an Olympus BX61microscope (Olympus) equipped with a motorised stage coupled to a computer running the stereology software Olympus VIS system composed by the Visiopharm Integrator System (VIS; Visiopharm, Horsholm, Denmark) software and the NewCast module for stereology acquisition and image analysis. We randomly overlayed a 2D grid of crosses on top of each section image (20,000 µm^2^ per cross) and the number of hitting crosses was used to obtain an unbiased estimate of the areas. The white matter area was relativised to the total area of each section.

#### 4.1.7. Motor Recovery Evaluation

The motor function recovery rate was assessed in an open field weekly for up to 4 weeks after injury following the Basso Mouse Scale (BMS) [34]. Evaluations were performed by two observers blinded to the treatment. Each feature of the scale was recorded and used to compute the BMS score and subscore as proposed by Basso and colleagues [34].

### 4.2. In Vitro

#### 4.2.1. Generation and Culture of SH-SY5Y^pcDNA3^ and SH-SY5Y^XIAP+^ Cell Lines

The hXIAP gene was amplified by PCR from a human cDNA library using the Moloney leukaemia virus transcriptase (Invitrogen, Waltham, MA, USA) and the primers listed in Appendix A. PCR was performed in a thermocycler (Bio-Rad, Hercules, CA, USA) programmed to complete 30 cycles of a three-step amplification (94 °C for 30 s, 60 °C for 30 s, and 72 °C for 1 min). The amplified gene was then cloned between the EcoRI and XhoI restriction sites of the pcDNA3 eukaryotic expression vector (Invitrogen) after the c-myc sequence, generating a c-myc + hXIAP fusion protein, important to be able to differentiate the expression of hXIAP transgene from endogenous XIAP protein. Ligation was performed by the T4 DNA-ligase and amplified in *E.coli* DH5 competent cells. The right positioning was confirmed by sequenciation.

For the generation of the cell line that stably overexpresses XIAP, the SH-SY5Y human neuroblastoma cell line was obtained from ATCC (see details in Appendix A) and cultured in an “SH-medium” composed of a 1:1 mixture of Eagle’s Minimal Essential Medium (Lonza, Basel, Switzerland) and Ham’s F12 medium (Lonza) supplemented with 10% fetal bovine serum (Lonza), 1% pyruvate (Lonza), and 1% Non-Essential-Amino-Acids (Lonza) at 37 °C and 5% CO_2_. Cells were plated in 100 mm^2^ culture dishes until reaching 70% confluence. XIAP-overexpressing SH-SY5Y cell lines were established by Transfectin (BioRad)-mediated transfection of the pcDNA3 + XIAP (SH-SY5Y^XIAP+^) or the empty pcDNA3 vectors (SH-SY5Y^pcDNA3^). After 48 h, transfected cells were selected through the gentamicin resistance cassette included in the pcDNA3 vector by incubation in a selection media composed of “SH-medium” supplemented with the gentamicin analogue geneticin G418 (700 μg/mL, Sigma-Aldrich). The medium was changed every three days. XIAP-overexpressing cell lines were established by clonal growth expansion. Finally, XIAP overexpression was confirmed by immunoblot.

#### 4.2.2. Immunoblot

After pro-apoptotic treatment (25 nM staurosporine (Sts) for 24 h), cells were detached from the plate by trypsin-EDTA (Lonza) treatment and lysed in buffer containing 4-(2-hydroxyethyl)-1-piperazineethanesulfonic acid (25 mM; HEPES, MERK), sodium chloride (150 mM; NaCl; Sigma-Aldrich), nonyl phenoxypolyethoxylethanol (1%; Tergitol-type NP40 surfactant, Sigma-Aldrich), sodium deoxycholate (1%; DOC, Sigma-Aldrich), glycerol (10%, USB Corporation, Affymetrix, Santa Clara, CA, USA), magnesium chloride(10 mM; MgCl_2_, Sigma-Aldrich), 3-[(3-cholamidopropyl)-di-methylammonium]-1-propane sulfonate (2 mM; CHAPS, Sigma-Aldrich), sodium dodecyl sulphate (0.1%; SDS, Sigma-Aldrich), ethylenediaminetetraacetic acid (2 mM; EDTA; USB Corporation, Affymetrix), and protease inhibitors. Homogenate containing 50µg of protein was resolved using conventional SDS–PAGE and immunoblotted as stated above (see antibodies in Appendix A).

#### 4.2.3. Cell Death Assay and Flow Cytometry

Both SH-SY5Y^pcDNA3^ and SH-SY5Y^XIAP+^ cell lines were grown in 100 mm^2^ culture dishes and, after reaching 80% confluence, were treated with thapsigargin, a pro-apoptotic drug, through the induction of endoplasmic reticulum stress (2.5 µM; TG Sigma-Aldrich) or Sts (5 to 25 nM) for 24 h.

For early death evaluation, we used the DY634 Annexin apoptosis detection kit (Immunostep, Salamanca, Spain) that measured the amount of phosphatidylserine transposition in the cell membrane right after a pro-apoptotic stimulus. After detachment with trypsin-EDTA (1×) and centrifugation (800× *g* for 3 min), cells were stained following the kit’s instructions. Briefly, cells were incubated in the kit’s binding buffer at 4 °C and then we added the annexin V dye (1:1100) at room temperature (RT) for 15 min in the dark.

For late death evaluation, cells were detached with trypsin-EDTA (1×), centrifuged (800× *g* for 3 min), washed with PBS, and fixed in 70% ethanol at −20 °C overnight. Then, the nuclei were stained with DAPI (1:3000) for 15 min.

After both staining protocols, cell death was quantified by the determination of the percentage of the population in the sub-G_0_ phase of the cell cycle. A dot plot showing pulse width versus area was used to distinguish between single cells and aggregates. A total of 10,000 gated single events were collected using a FACS Canto II flow cytometer (BD Biosciences, Franklin Lakes, NJ, USA). Data were analysed with the FACS Diva 6.1software (BD Biosciences) and analysed with Flow Jo Software (Celeza GmbH, Olten, Switzerland).

#### 4.2.4. Evaluation of CASP3/7 Activity

The activities of the effector caspases-3 and -7 were measured with the Caspase-Glo 3/7 Assay (Promega, Fitchburg, WI, USA). Briefly, both SH-SY5Y cell lines were plated in 96-well plates with 100 µL of culture medium as described above. After 24 h in culture, cells were treated with Sts (25 nM) for 24 h. Following treatment, 100 µL of the Caspase-Glo reagent was added to each well, and luminescence was measured every 5 min for 60 min in a plate reader luminometer (Infinite M200, Tecan Group Ltd. Mannendorf, Switzerland).

### 4.3. Data Analysis

All data are expressed as means ± SEM as indicated in figure legends. Statistical significance of the treatment effects was tested using the paired or non-paired Student’s *t*-test, the analysis of variance test (ANOVA), or the Chi-square test depending on the characteristics of the data. Normality and homoscedasticity of the data were assessed using Shapiro–Wilk and Bartlett tests, respectively, using the Shapiro.test and Bartlett.test functions of R software [40]. Statistical analyses and graphics were carried out and made using Prism Software 5.0 (GraphPad Software Inc., Insight Partners, New York, NY, USA) and R statistical language. Differences were considered statistically significant when the *p*-value was <0.05.

## 5. Conclusions

In summary, we confirm here the crucial role of XIAP in the experimental SCI model, supporting the view that an increase in XIAP expression in the spinal cord cells, specifically in neurons, can be an effective strategy to ameliorate deficits related to SCI. As shown in previous work and as extended here with a focus on neurons, a reduction in the levels of XIAP in the spinal cord neurons is one contributing factor for the activation of the apoptotic program leading to neuron death during the secondary damage after SCI. Strategies targeting XIAP in neuronal cells in the spinal cord may therefore yield promising therapeutic tools for SCI recovery, by protecting neural cells from secondary deleterious effects. Such approaches could even be combined with systemic pharmacological treatment using different drugs including ucf-101. In the long run, this knowledge may help to preserve and re-establish motor function, partly through the preservation of local networks, and concomitantly lead to a significant improvement in the quality of life of SCI patients.

## Figures and Tables

**Figure 1 ijms-24-02791-f001:**
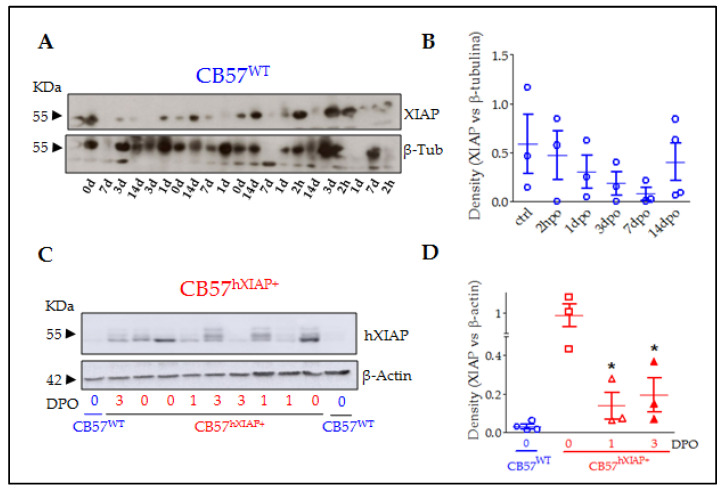
XIAP expression in the spinal cord of CB57^WT^ and CB57^hXIAP+^ mice was reduced after SCI. Representative immunoblot of XIAP levels in CB57^WT^ (**A**) and hXIAP in both CB57^WT^ and CB57^hXIAP+^ (**C**) mouse spinal cord samples. Band densitometry (relativised to the loading control) reveals that SCI induces a decrease in endogenous XIAP and human hXIAP expression in CB57^WT^ (**B**) and CB57^hXIAP+^ (**D**) mice, respectively (* = *p*-value < 0.05 in Student’s *t*-test *n* = 3–4 animals/group).

**Figure 2 ijms-24-02791-f002:**
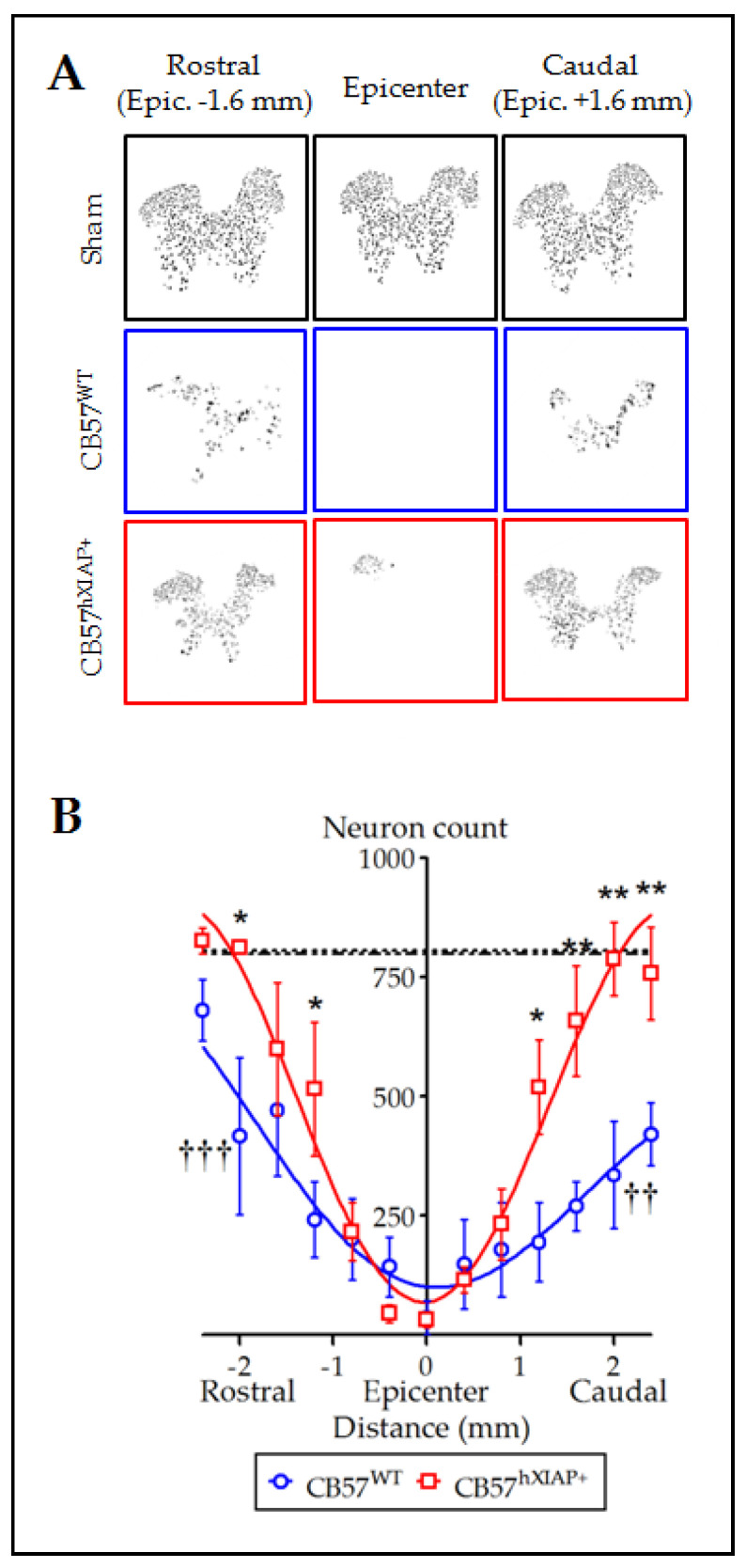
Overexpression of hXIAP reduces neuronal death after SCI. (**A**) Representative pictures obtained after processing of stained sections with TruAI integrated cellSens Dimensions software for neuron counting. (**B**) Estimations of the number of neurons in transverse sections comprising 4.8 mm surrounding the injury epicentre of the spinal cord show that CB57^hXIAP+^ mouse sections (red symbols) have a significant increase in neuron counts with respect to CB57^WT^ mouse sections (blue symbols), particularly in sections caudal to the injury epicentre (symbols represent mean ± SEM and line the polynomial non-linear regression adjusts; * = *p*-value < 0.05 and ** = *p*-value < 0.01 for injured CB57^hXIAP+^ vs. CB57^WT^ mice; in Student’s *t*-test; †† = *p*-value < 0.01 and ††† = *p*-value < 0.001 for injured CB57^WT^ mice vs. sham animals (dash-dotted line = sham CB57^WT^; dotted line = CB57^hXIAP+^); *n* = 3–6 animals/group).

**Figure 3 ijms-24-02791-f003:**
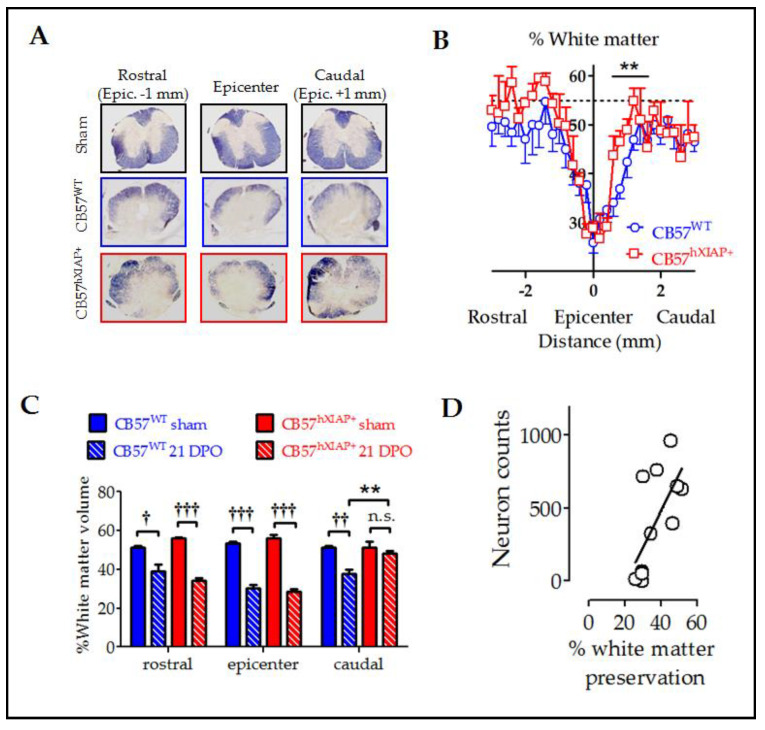
Reduction in histological damage at 21 dpo by hXIAP overexpression in CB57^hXIAP^ mice. (**A**) Representative images of spinal cord sections from sham (black), 21 dpo injured CB57^WT^ mice (blue), and CB57^hXIAP+^ 21 dpo injured mice (red). (**B**) Estimations of the percentage of preserved white matter in transverse sections comprising 3 mm surrounding the injury epicentre of the spinal cord show that CB57^hXIAP+^ mice (red symbols) slices have a significant increase in spared tissue area in sections caudal to the injury epicentre compared with CB57^WT^ mice (blue symbols) (symbols represent mean ± SEM; ** = *p*-value < 0.01 in Student’s *t*-test vs. CB57^WT^ mice.; *n* = 4–6 animals/group). (**C**) Estimations of the volume of spared white matter in cylinders 300 µm-long centred at the injury epicentre, as well as caudal and rostral to this region, also reveal a protective effect of hXIAP overexpression (red bars) in the region caudal to the injury zone in comparison with WT mice (blue bars) (bars represent means ± SEM; † = *p*-value < 0.05, †† = *p*-value < 0.01, ††† = *p*-value < 0.001 vs. their correspondent sham samples for each mouse strand; ** = *p*-value < 0.01, in Student’s *t*-test, *n* = 4–6 animals/group). (**D**) Dot-plot correlating preserved white matter volumes with the corresponding neuron counts for each section (Figure 2) at the SCI epicentre and at both caudal and rostral directions of transgenic and WT mice altogether. Pearson correlation coefficient = 0.6, *p*-value = 0.012.

**Figure 4 ijms-24-02791-f004:**
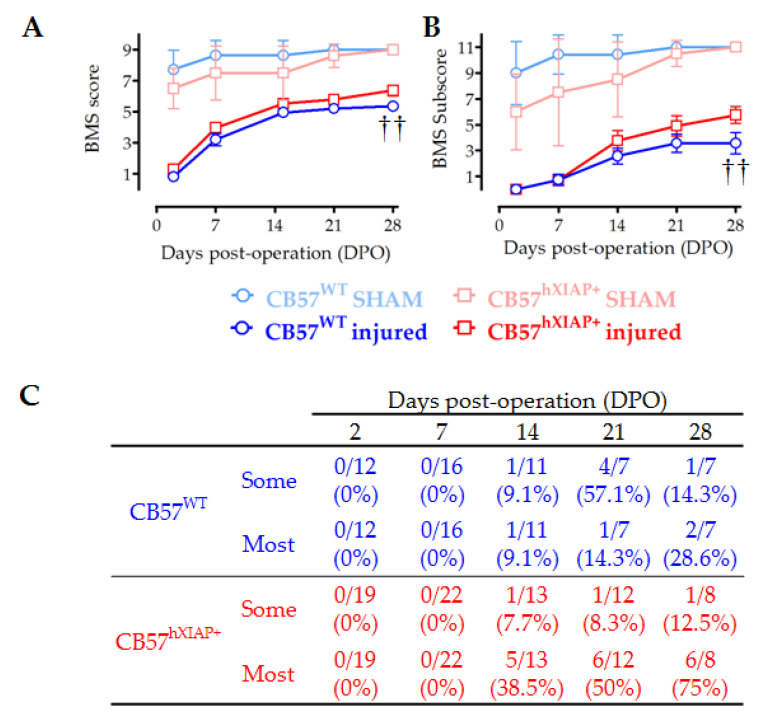
Improvement in locomotor capacities by overexpression of hXIAP after SCI. We used BMS to determine the locomotor skills of spinal cord-injured WT (CB57^WT^, dark blue line and symbols are for injured mice and light blue line and symbols are for sham animals) and hXIAP overexpressing mice (CB57^hXIAP+^, red line and symbols are for injured mice and light red and symbols are for sham animals). (**A**) BMS score reveals that overexpression of hXIAP reduces the locomotor impairment derived from SCI (†† = *p*-value < 0.01 for the effect of genetic background in a two-way ANOVA). (**B**) Improvements become more evident when locomotion parameters are coded according to the BMS subscore (†† = *p*-value < 0.01 for the effect of genetic background in a two-way ANOVA). (**C**) The specific analysis of interlimb coordination shows that overexpression of hXIAP tends to increase the percentage of animals with coordinative capacities at 21 and 28 dpo (symbols represent mean ± SEM).

**Figure 5 ijms-24-02791-f005:**
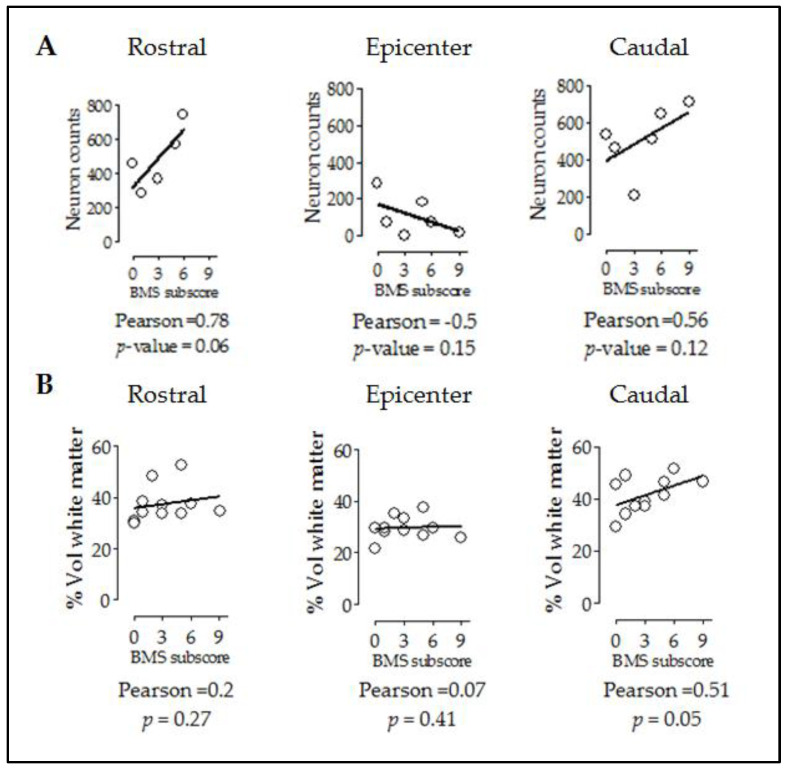
Improvement in locomotor capacities by overexpression of hXIAP after SCI correlates with both neuron counts and tissue preservation levels. Dot-plot correlating BMS subscores with neuron counts (**A**) and preserved white matter volumes (**B**) at different distances from the injury site. Lines represent the correlation fit of both CB57^WT^ and CB57^hXIAP+^ animal data altogether.

**Figure 6 ijms-24-02791-f006:**
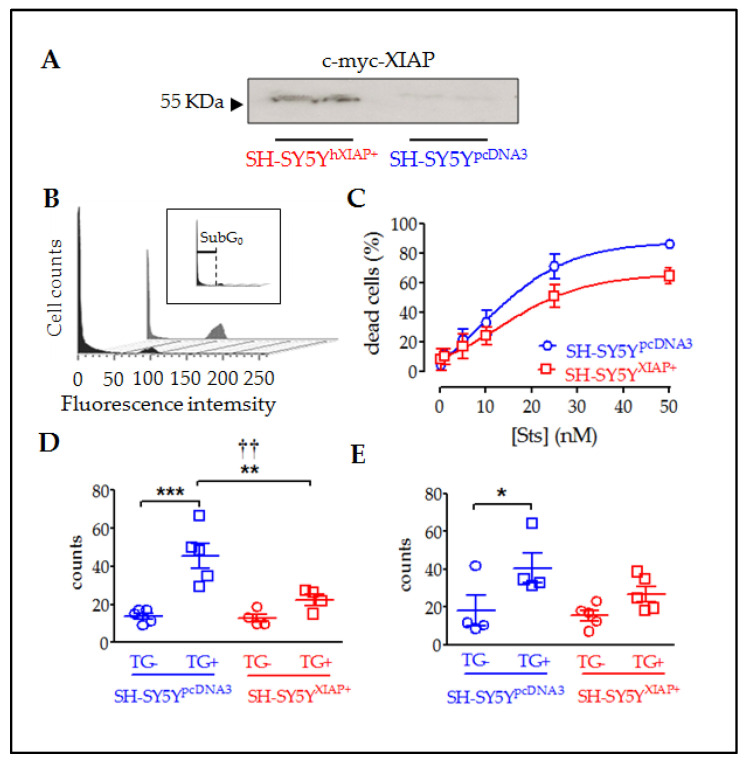
Reduction in cell death in SH-SY5Y cultures overexpressing XIAP. (**A**) Immunoblot showing the overexpression of c-myc-XIAP fusion protein in homogenates obtained from SH-SY5Y^pcDNA3^ and SH-SY5Y^XIAP+^ cells. (**B**) Representative flow cytometry histograms of DAPI-stained SH-SY5Y cell cycle under Sts stimulation. The black profile corresponds to SH-SY5Y^WT^-treated control cultures and the grey profile to SH-SY5Y^XIAP+^-treated cells. Insert in B shows the sub-Go region (apoptotic cells with condensed nuclei) of the cell cycle used to evaluate the percentage of cell death in the population. (**C**) Dose–response of the percentages of apoptotic cells in the SH-SY5Y^WT^ (blue symbols) and SH-SY5Y^XIAP+^ (red symbols) cell cultures after treatment with increasing concentrations of Sts (symbols represent the mean ± SEM and lines the Michaelis–Menten non-linear fit of the data; ** = *p*-value < 0.01 in paired Student’s *t*-test; † = *p*-value < 0.05 in two-way ANOVA of genetic background effect). Early apoptotic death was measured through annexin V staining (**D**) and late cell death was measured by DAPI staining (**E**) after TG treatment in SH-SY5Y^pcDNA3^ (blue symbols) and SH-SY5Y^XIAP+^ (red symbols) cell cultures (dot plots represent mean ± SEM; *** = *p*-value < 0.001, ** = *p*-value < 0.01, and * = *p*-value < 0.05 in non-paired Student’s *t*-test. †† = *p*-value = 0.01 in two-way ANOVA of genetic background effect; *n* = 4–5 independent experiments).

**Figure 7 ijms-24-02791-f007:**
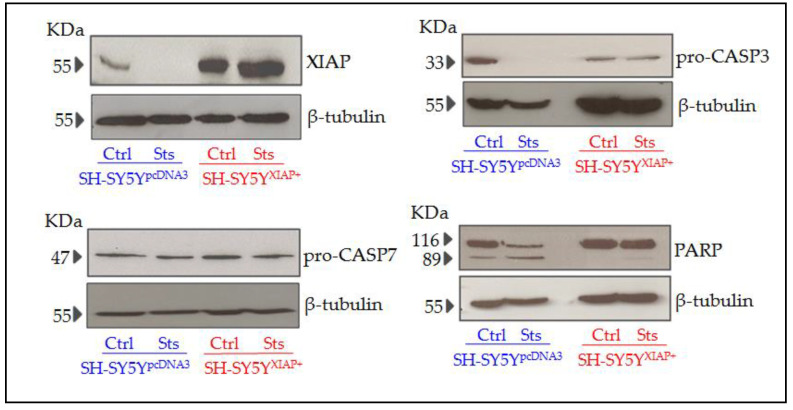
Expression levels of different apoptotic-related proteins in SH-SY5Y cells after Sts treatment. Representative immunoblots of XIAP, pro-CASP3, PARP, and pro-CASP7 and loading control β-tubulin proteins expression in SH-SY5Y^pcDNA3^ and SH-SY5Y^XIAP+^ cell line lysates before and after treatment with Sts (25 nM for 24 h).

**Figure 8 ijms-24-02791-f008:**
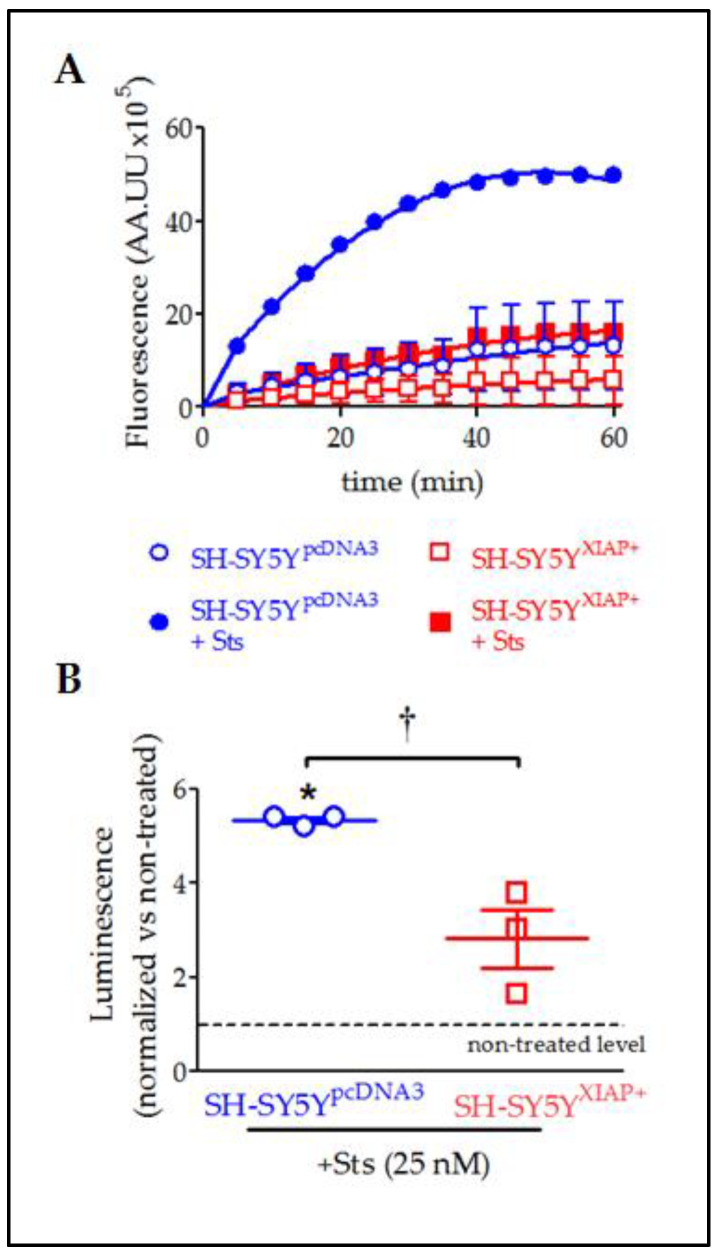
Reduction in Sts-induced activation of CASP3/7 by overexpression of XIAP in SH-SY5Y cells. (**A**) Kinetics of CASP3/7 activity in both SH-SY5Y^pcDNA3^ (blue symbols) and SH-SY5Y^XIAP+^ cells (red symbols), treated at time 0 with Sts (filled symbols) or vehicle (empty symbols). (**B**) The dot plot summarises the Sts effect in CASP3/7 activity for SH-SY5Y^pcDNA3^ (blue symbols) and SH-SY5Y^XIAP+^ cells (red symbols). Data were normalised by non-treated levels (dot line) after 30 min (dot plot represents means ± SEM; * = *p*-value < 0.05 in paired Student’s *t*-test for Sts-treated SH-SY5Y^pcDNA3^ vs. vehicle-treated cells; † = *p*-value < 0.05 in a two-way ANOVA of genetic background effect; *n* = 3 independent experiments).

## Data Availability

Not applicable.

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
