# Peer review of "Overexpression of the X-Linked Inhibitor of Apoptosis Protein (XIAP) in Neurons Improves Cell Survival and the Functional Outcome after Traumatic Spinal Cord Injury"

_ijms, 2023, doi:10.3390/ijms24032791_

Round 1

Reviewer 1 Report

I consider that this study is well designed, performed and summarized. I have one comment about the usage of words, "traumatic" and "trauma". These words are used in the title and introduction but their meanings are not clear. What's the difference between "traumatic spinal cord injury" and "spinal cord injury"?  "traumatic spinal cord injury" sounds like it has psychological damage and "spinal cord injury" does not. Also what is the difference between "spinal cord injury" and spinal cord trauma"? These words should be clarified.

Author Response

Reviewer #1:

I consider that this study is well designed, performed, and summarized. I have one comment about the usage of the words, "traumatic" and "trauma". These words are used in the title and introduction but their meanings are not clear. What's the difference between "traumatic spinal cord injury" and "spinal cord injury"?  "traumatic spinal cord injury" sounds like it has psychological damage and "spinal cord injury" does not. Also, what is the difference between "spinal cord injury" and spinal cord trauma"? These words should be clarified.

Answer to Reviewer #1:

Dear reviewer, thanks for your comments. Regarding your question, we have tried to clarify the concepts of trauma and traumatic spinal cord injury as you mention. Traumatic spinal cord injury refers (as traumatic brain injury) to the condition resulting from mechanical damage to the spinal cord. The term traumatic SCI differentiates this etiology from non-traumatic origins such as diabetes, cancer, or degenerative processes. To make it more straightforward and to differentiate mechanical damage from any psychological consequence of  SCI we have added the word mechanical before trauma in both the abstract and in paragraph two of the introduction. We have also added a brief explanation of the mechanical origin of traumatic SCI in the first sentence of the introduction:

"Mechanical damage to the spinal cord results in a condition termed traumatic spinal cord injury (tSCI), which is characterized by the partial or total loss of ..."

Reviewer 2 Report

Thank you for giving me the chance to review the manuscript entitled" Overexpression of the X-linked Inhibitor of Apoptosis Protein (XIAP) in Neurons Improves Cell Survival and the Functional Outcome after Traumatic Spinal Cord Injury ". In the present study, the author proved that the levels of XIAP expression are an important factor for the outcome after spinal cord trauma and identifies XIAP as an important therapeutic target for alleviating the deleterious effects of SCI. Overall the paper is written well and clearly. 

1.There are too many irrelevant contents about spinal cord injury in the introduction , which need to be further optimized.

2.Can you express more clearly about the neurons in the center of injured spinal cord between different groups?

3.As shown in Figure3, in addition to statistical analysis results, the original tissue picture should be provided.

4.As shown in Figure4, without Sham operation group.

5.In vitro experiment, why not use primary cultured neuron cells from transgenic mouse spinal cord for related experiments?

6.Some references are old and need to be updated.

Author Response

Answer to Reviewer #2:

Thank you for giving me the chance to review the manuscript entitled" Overexpression of the X-linked Inhibitor of Apoptosis Protein (XIAP) in Neurons Improves Cell Survival and the Functional Outcome after Traumatic Spinal Cord Injury ". In the present study, the author proved that the levels of XIAP expression are an important factor for the outcome after spinal cord trauma and identifies XIAP as an important therapeutic target for alleviating the deleterious effects of SCI. Overall the paper is written well and clearly.

  1. There are too much irrelevant contents about spinal cord injury in the introduction, which need to be further optimized.

Dear reviewer, thanks for your comments and suggestions which greatly improve the manuscript. Concerning this first point, we have followed your suggestion. The first part of the introduction has been reduced to the essential information about the relevance of SCI, deleting unnecessary references too. The text is presented as follows:

(Page 1, paragraph 1)

" Mechanical damage to the spinal cord results in a condition termed traumatic spinal cord injury (tSCI) which is characterized by partial or total loss of sensory-motor and autonomic functions below the injury level. It is one of the leading causes of disability in developed countries with big personal, economic, familiar, and social impact [1–3]. Several promising therapies have been developed during the last three decades [3], some in different stages of clinical trials [3, 4], but only early surgical decompression is routinely applied in clinical practice [5].

  1. Can you express more clearly the neurons in the center of the injured spinal cord between different groups?

Thanks for your comment. Reviewing the manuscript, we have seen that due to a typographic mistake, we wrote that the neuron counts were made at 1 and 3 DPO, but all histological analysis was made at 21 DPO as mentioned in the Methods chapter. We have corrected this mistake and completed the paragraph in section 2.2 of the results to provide the neuron counts in the epicenter area in sham and at 21 DPO conditions in both mouse strings. The sentence (Page 3, heading 2.2, paragraph 1) now indicates that:

" These analyses revealed that at 21 DPO virtually all neurons have disappeared within the 400 µm surrounding the injury epicenter in both CB57WT and CB57hXIAP animals (number of neurons in CB57WT mice = 36.3 ± 35.1 vs. SHAM = 800 ± 39; and in CB57hXIAP+ mice = 32.75 ± 15.6 vs. SHAM = 807.7 ± 72.1)".

  1. As shown in Figure 3, in addition to statistical analysis results, the original tissue picture should be provided".

We have modified Figure 3 to include a panel with representative images of spinal cord sections stained with the eriochrome cyanide method (Figure 3A, page 5). These images are part of the full set used for the quantification of spared white matter in each condition and mouse string, in the epicenter and at 1 mm at both caudal and rostral directions. So, old panels A, B, and C have become now panels B, C, and D in the figure, figure legend, and in the main text. We also changed the Legend for Figure 3 to describe the new panel 3A:

(Page 5, Figure 3 Legend)

"Figure 3. Reduction of histological damage at 21 DPO by hXIAP overexpression in CB57hXIAP mice. (A) Representative images of spinal cord sections from sham (black), 21 DPO injured CB57WT mice (blue), and CB57hXIAP + 21 DPO injured mice (red). (B) Estimations of the percentage of preserved white matter in transverse sections comprising 3 mm surrounding the injury epicenter of the spinal cord show that CB57hXIAP+ mice (red symbols) slices have a significant increase of spared tissue area in sections caudal to the injury epicenter compared with CB57WT mice (blue symbols) (Symbols represent mean ± SEM; ** = p-value < 0.01 in Student's t-test vs. CB57WT mice.; n = 4-6 animals/group). (C) Estimations of the volume of spared white matter in cylinders of 300 µm long centered at the injury epicenter, as well as caudal and rostral to this region, also reveal a protective effect of hXIAP overexpression (red bars) in the region caudal to the injury zone in comparison with WT mice (blue bars) (Bars represent means ± SEM; † = p-value < 0.05, †† = p-value < 0.01, ††† p-value < 0.001 vs. their correspondent sham samples for each mice strand; ** = p-value < 0.01, in Student's t-test, n = 4-6 animals/group). (D) Dot-plot correlating preserved white matter volumes with the corresponding neuron counts for each section (Figure 2) at the SCI epicenter and at both caudal and rostral directions of transgenic and WT mice altogether. Pearson correlation coefficient = 0.6, p = 0.012.".

  1. As shown in Figure 4, without Sham operation group.

We have changed Figures 4A and 4B adding the data of BMS score and subscore of sham animals for both WT and transgenic mice (light-colored lines and symbols). Accordingly, we have changed the Legend for Figure 4 as follows:

(Page 6, Figure 4 Legend)

"Figure 4. Improvement of locomotor capacities by overexpression of hXIAP after SCI. We used BMS to determine the locomotor skills of spinal cord-injured WT (CB57WT,. dark blue line and symbols for injured and light blue line and symbols for sham animals) and hXIAP overexpressing mice (CB57hXIAP+, red line and symbols for injured and light red and symbols for sham animals). (A) BMS score reveals that overexpression of hXIAP reduces the locomotor impairment derived from SCI (†† = p-value < 0.01 for the effect of genetic background in a two-way ANOVA). (B) Improvements become more evident when locomotion parameters are coded according to the BMS subscore (†† = p-value < 0.01 for the effect of genetic background in a two-way ANOVA). (C) The specific analysis of interlimb coordination shows that overexpression of hXIAP tends to increase the percentage of animals with coordinative capacities at 21 and 28 DPO (Symbols represent mean ± SEM)".

  1. In vitro experiment, why not use primary cultured neuron cells from transgenic mouse spinal cord for related experiments?

Thanks for the idea but, unfortunately, we had a very limited colony (already extinguished) of XIAP transgenic mice that we employed to carry out as many in vivo analyses as possible. Indeed, we considered culturing adult spinal neurons but we discarded this possibility because it would require too many individuals from the colony due to the very low efficiency of these cultures. However, this approach as well as others such as single cell analysis of the spinal cord from transgenic mice would be very interesting for future studies as we have indicated in the third paragraph of the discussion (page 6):

“Further studies, including cultures of spinal neurons overexpressing XIAP or single cell analyses of CB57hXIAP+ mice, would be required to define the precise survival/cell death pathways that become regulated by XIAP overexpression in the specific neuronal populations of the spinal cord.”

  1. Some references are old and need to be updated.

We reviewed the text and updated the cited literature as much as we could. Where we considered appropriate old references have been replaced by updated literature. Some other references have been directly deleted due to the reduction of the first part of the introduction or because they are redundant or unnecessary. However, some old references have been conserved due to their importance, because they are the primary source of information about a protein or a mechanism, or even cited by more modern references but without any additional information. Although they are relatively old, we think that is important to maintain these references in the text. Additions, deletions, or replacements of references are indicated in the main text.